# Evaporated nanometer chalcogenide films for scalable high-performance complementary electronics

Ao Liu [1], Huihui Zhu [1], Taoyu Zou[1], Youjin Reo [1], Gi-Seong Ryu[1] & Yong-Young Noh [1] ✉

The exploration of stable and high-mobility semiconductors that can be grown over a large area using cost-effective methods continues to attract the interest of the electronics community. However, many mainstream candidates are challenged by scarce and expensive components, manufacturing costs, low stability, and limitations of large-area growth. Herein, we report wafer-scale ultrathin (metal) chalcogenide semiconductors for high-performance complementary electronics using standard room temperature thermal evaporation. The n-type bismuth sulfide delivers an in-situ transition from a conductor to a high-mobility semiconductor after mild post-annealing with self-assembly phase conversion, achieving thin-film transistors with mobilities of over 10 cm$^2$ V$^{-1}$ s$^{-1}$, on/off current ratios exceeding 10$^8$, and high stability. Complementary inverters are constructed in combination with p-channel tellurium device with hole mobilities of over 50 cm$^2$ V$^{-1}$ s$^{-1}$, delivering remarkable voltage transfer characteristics with a high gain of 200. This work has laid the foundation for depositing scalable electronics in a simple and cost-effective manner, which is compatible with monolithic integration with commercial products such as organic light-emitting diodes.

Thin-film transistor (TFT) technology has promoted the rapid development of inexpensive and large flat-panel displays. It is being applied widely in microprocessors, sensors, radio-frequency identification tags, wearable electronics, and other Internet-of-Things devices[1-7]. Unlike state-of-the-art high-performance silicon metal-oxide-semiconductor field-effect transistors (which involve stringent process restrictions), TFT technology is unique in that it can be manufactured with high yield on various large-area substrates via cost-effective process[4]. Decades of research has been devoted to the examination of TFT channel semiconductors including metal chalcogenides/oxides/halides, organics, and low-dimensional nanomaterials[7-11]. Metal oxides and chalcogenides show high electrical performance and stability. However, the use of expensive components (e.g., In and Ga) and toxic metals such as Cd and Pb involve high manufacturing costs and hazards to environmental safety. Notwithstanding the potential functionalities of emerging low-

dimensional nanomaterials, it is difficult to achieve wafer-scale homogeneous deposition via an inexpensive and high reproducible way, limiting their application in large-area TFTs[12].

Among the various thin-film deposition methods for large-area electronics, thermal evaporation provides remarkable deposition scalability and reproducibility, and enables precise film thickness control, homogeneous deposition on textured substrates, and highly customized multilayer stack growth. The commercialization of eighth generation (2200 × 2500 mm) organic light-emitting diodes (OLED) using thermal evaporation shows the feasibility of mass production of TFTs. In addition, film patterning with a size of several tens of micrometers can be achieved conveniently over a large area using fine metal masks. Among potential semiconductors capable of thermal evaporation, bismuth sulfide (Bi$_2$S$_3$) shows great potential for TFT channels due to its eco-friendliness and cost-effectiveness in conjunction

[1]Department of Chemical Engineering, Pohang University of Science and Technology, 77 Cheongam-Ro, Nam-Gu, Pohang 37673, Republic of Korea.
✉e-mail: yynoh@postech.ac.kr

with high electron mobilities of up to 640 cm$^2$ V$^{-1}$ s$^{-1}$ [13–16]. Bi$_2$S$_3$ has been studied extensively in photoelectric and thermoelectric devices benefiting from the suitable optical bandgap of 1.3–1.7 eV and high heat/electricity conversion[13,17–20]. As a p-type candidate, tellurium (Te) has recently attracted substantial attention for high-performance TFT fabrication owing to its high hole mobility and stability[21–23]. In the efforts to deposit (metal)chalcogenide thin films over a large area in a cost-effective manner, the main attention has been paid to the solution process[24–27]. However, the strong covalent bonding of these solids requires complex and toxic synthetic processes and has been challenging to integrate with the conventional complementary metal-oxide-semiconductor (CMOS) technology. Another noteworthy solution approach is liquid-phase exfoliation. It is used widely to obtain layered two-dimensional (2D) metal chalcogenide nanosheets[7,28,29]. Recent studies achieved high-performance molybdenum sulfide (MoS$_2$) TFTs with an electron mobility ($\mu_e$) of up to ~10 cm$^2$ V$^{-1}$ s$^{-1}$ and on/off current ratio ($I_{on}/I_{off}$) of 10$^4$–10$^6$ by controlling the defect healing and intercalation chemistry[28,30]. However, the structural surface defects from the high-energy exfoliation process, non-uniform number of layers in each nanoflake, and inter-flake electrical resistance may limit large-area uniformity and further performance optimization[28].

In this Article, we report sub-nanometer (metal)chalcogenide thin films deposited over large areas through industry-compatible room temperature (RT) thermal evaporation and explore their applications in TFTs and complementary electronics. Aided by the high vapor pressure of (metal)chalcogenides, thermally evaporated Bi$_2$S$_3$ exhibits uniform films over large areas with nanometer-level precise thickness control. The as-deposited Bi$_2$S$_3$ provides an amorphous structure containing sulfur-rich components with conductor-like behavior. Mild post-annealing can modulate the composition and drive self-assembly crystallization with the conversion to a high-mobility stable channel for TFT with $\mu_e$ of over 10 cm$^2$ V$^{-1}$ s$^{-1}$ and $I_{on}/I_{off}$ exceeding 10$^8$. The high-gain complementary inverter is further realized with the high-mobility p-channel Te TFT, indicating the great potential of thermally evaporated (metal)chalcogenides for large-area CMOS circuit integration.

## Results

### Electrical characterizations of thermally-evaporated Bi$_2$S$_3$ TFTs

The Bi$_2$S$_3$ channel layers were deposited by RT thermal evaporation on atomic layer deposition HfO$_2$ (40 nm)/p$^+$-Si substrates. This was followed by post-annealing at 200–300 °C for 30 min (see the Methods section for further details). The gold source/drain electrodes were then deposited on the patterned Bi$_2$S$_3$ to construct bottom-gate top-contact TFTs (Fig. 1a). Typical transfer curves of Bi$_2$S$_3$ TFTs are shown in Fig. 1b. The TFTs with the as-evaporated Bi$_2$S$_3$ channel exhibited conductor-like behavior with a constant source–drain current ($I_{DS}$) of ~1 mA. Such characteristics are generally caused by the excessive concentration of charge carriers in the channel layer and thus, the negligible gate bias modulation capability. It is noteworthy that mild post-annealing achieved the significant n-channel transistor characteristic with the desired enhancement operation mode (threshold voltage, $V_{TH}$ > 0 V). The improved $\mu_e$ at higher annealing temperatures can be attributed to the enhanced long-range ordering of the microstructures. Among these, the post-annealing at 250 °C yielded a well-balanced TFT performance, including a high $\mu_e$ of 12.5 cm$^2$ V$^{-1}$ s$^{-1}$, high $I_{on}/I_{off}$ of 2 × 10$^8$, $V_{TH}$ of 1.1 V, subthreshold swing (SS) of 0.2 V dec$^{-1}$, small hysteresis <1 V, and high stabilities (Fig. 1c, Supplementary Fig. 1). The corresponding output curves show significant current linearity at low source–drain voltages, indicating the Ohmic contact between Bi$_2$S$_3$ channel and Au electrodes (Fig. 1d). The transmission-line method[31] was employed to evaluate the contact resistance ($R_c$) and it was calculated to be 360 Ω cm for the 250 °C-Bi$_2$S$_3$ TFT (Fig. 1e). We then performed temperature-dependent measurements to investigate the charge transport properties of the optimized 250 °C-Bi$_2$S$_3$ channel (Fig. 1f). The TFT mobilities first increased as the temperature

decreased from 293 to 223 K. This is a typical band-like transport commonly observed in highly crystalline and high mobility semiconductors[32–35]. Therefore, we could infer a high degree of order in post-annealed Bi$_2$S$_3$ thin films. When the measurement temperatures decreased further, the electron transport became thermally activated. This was dominated by shallow traps in Bi$_2$S$_3$, which caused a marginal reduction in mobility. Such temperature-dependent characteristics differ from the observations of solution-based liquid-phase exfoliated metal chalcogenide TFTs in that the thermal activation is governed over the temperature range, which is likely to be associated with activated interflake hopping[36,37].

In addition to facile post-annealing, the Bi$_2$S$_3$ channel thickness had a significant effect on the key TFT parameters, e.g., $\mu_e$ and $I_{on}/I_{off}$ (Fig. 1g, Supplementary Fig. 2). A suitable Bi$_2$S$_3$ thickness of 5 nm was used for the above device characterization. When the Bi$_2$S$_3$ channel was downscaled to 3 nm, the remarkable electrical performance was maintained with a similarly high $I_{on}/I_{off}$ of 10$^8$ and a marginally reduced $\mu_e$ of ~9 cm$^2$ V$^{-1}$ s$^{-1}$. During transistor operation, the accumulated charge carriers were confined to a narrow region (3–5 nm) close to the channel/dielectric interface. In channel layers that were excessively thin, the carrier transport could undergo backscattering owing to roughness, dangling bonds, and defects[21]. When the channel thicknesses exceeded 5 nm, $\mu_e$ increased monotonically from 20.2 to 34.5 cm$^2$ V$^{-1}$ s$^{-1}$ for the TFTs based on 8 and 12 nm Bi$_2$S$_3$ channel layers. A downward trend from 10$^5$ to 10$^3$ was observed for $I_{on}/I_{off}$. This can be interpreted as a reduced electrostatic control for TFTs based on thicker channel layers, which is commonly observed in different material systems[21,38,39]. Finally, to examine the scalability of thermally evaporated Bi$_2$S$_3$, we fabricated a TFT array on a 4 inch HfO$_2$/p$^+$-Si substrate and randomly measured 100 TFTs (Fig. 2h). The devices exhibited remarkable uniformity and a narrow performance distribution with $\mu_e$ in the range of 10.8–14.2 cm$^2$ V$^{-1}$ s$^{-1}$ and $I_{on}/I_{off}$ of 1–4 × 10$^8$ (Fig. 2i). It is noteworthy that such wafer-scale deposition of 5 nm Bi$_2$S$_3$ film requires only 40 mg of Bi$_2$S$_3$ powder (~USD 0.28), representing a significantly low material cost.

### Bi$_2$S$_3$ thin-film characterizations

A series of film analyses were performed to clarify the effects of post-annealing on the significantly different TFT performance. First, the crystal structure of the film was analyzed. Bi$_2$S$_3$ is theoretically composed of a lamellar structure with (Bi$_4$S$_6$)$_n$ ribbons stacked along the c-axis through van der Waals interactions (Fig. 2a)[13,40]. This atomic chain configuration ensures an intrinsically benign grain boundary and efficient charge transport[41]. Based on X-ray diffraction (XRD), the as-evaporated Bi$_2$S$_3$ film showed an evident amorphous characteristic, and a polycrystalline texture was observed after post-annealing at 250 °C (Fig. 2b). Cross-sectional high-resolution transmission electron microscopy (HRTEM) was performed to obtain more precise information regarding the microscopic crystal structure. As shown in Fig. 2c–e, the TEM image and selected area electron diffraction (SAED) pattern exhibit the typical amorphous structure of as-deposited Bi$_2$S$_3$ without a perceptible long-range order. This initial amorphous state is favorable to subsequent scalable growth owing to its superior uniformity. After the post-annealing at 250 °C, a well-defined laminar texture was observed without visible defects (Fig. 2f, g). A lattice spacing of 0.56 nm corresponding to the (200) crystalline plane of Bi$_2$S$_3$ was measured. It was also verified by the fast Fourier transform (FFT) spot patterns of the marked frames (Fig. 2h). The thickness of the Bi$_2$S$_3$ film (i.e., the number of layers) could be controlled precisely by manipulating the evaporation time and rate. It was noted that the efficient evaporation process enabled the deposition of a 5 nm Bi$_2$S$_3$ thin film in ~25 s (rate: ~2 Å s$^{-1}$). This provided a high throughput compared with other film deposition methods.

The optical spectra exhibited increased light absorption after the annealing at 250 °C, with the bandgap ($E_g$) marginally reduced from 1.6

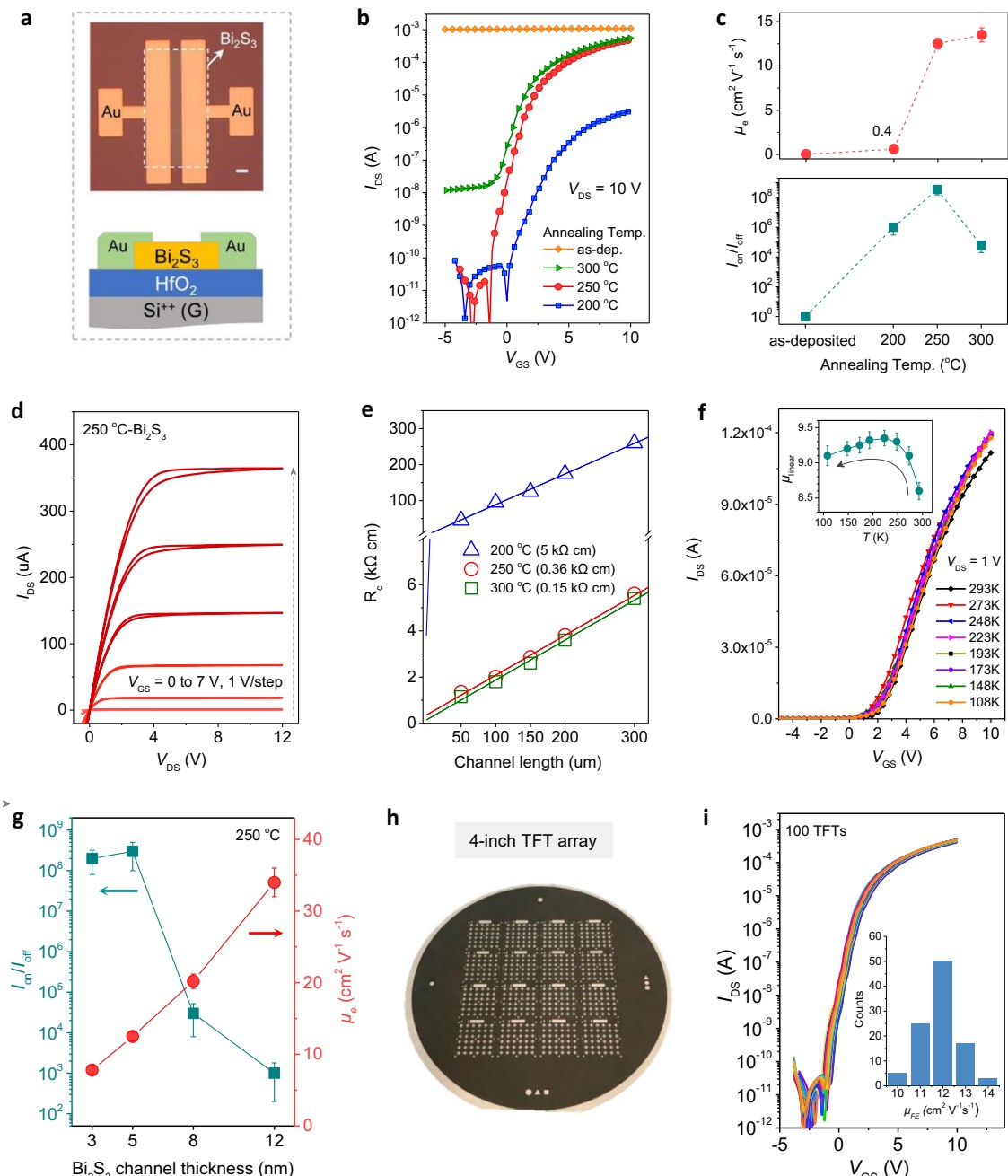

**Fig. 1 | Electrical characterizations of Bi₂S₃ TFTs. a** TFT structure used in this study and optical image of typical $Bi_2S_3$ TFT (scale bar: 100 μm). **b** Transfer curves of $Bi_2S_3$ TFTs as a function of post-annealing temperature. **c** Corresponding summarization of values of electron mobility ($\mu_e$) and $I_{on}/I_{off}$. The error bars were calculated from 10 individual TFTs. **d** Output curves of the optimized $Bi_2S_3$ TFT annealed at 250 °C. **e** Transmission-line method (TLM) plots with different channel lengths. Here, contact resistance ($R_c$) is determined from the y-axis intercept. **f** Transfer curves and mobility variations of the optimized 250 °C-$Bi_2S_3$ TFT as a function of the measurement temperature in vacuum. The error bars were obtained from 5 individual devices. **g** Summarization of $\mu_e$ and $I_{on}/I_{off}$ for 250 °C-$Bi_2S_3$ TFTs with different channel thicknesses. The error bars were obtained from 10 individual devices. **h** Optical image of TFT arrays on a 4-inch HfO₂/p⁺-Si substrate. **i** Transfer curves of 100 randomly measured 250 °C-$Bi_2S_3$ TFTs. The inset shows the statistical results of $\mu_e$.

to 1.5 eV (Fig. 3a). The atomic force microscope (AFM) images displayed a substantial uniformity and ultra-smooth surface morphology, with small root mean square (RMS) values of 0.28 and 0.24 nm for the as-deposited and 250 °C-$Bi_2S_3$ thin films, respectively (Fig. 3b, c). Such atomically smooth topography allows for a high-quality interface and high device yield over a large area. The typical polycrystalline texture is observed in the 250 °C-annealed film with the average grain size of hundred nanometers (Supplementary Fig. 3). We then performed X-ray photoelectron spectroscopy (XPS) to analyze the film components. For both as-deposited and 250 °C-annealed samples, the Bi 4$f$

spectra showed two peaks at 163.5 and 158.2 eV. These match closely with the characteristic $Bi^{3+}$ peaks in $Bi_2S_3$ (Supplementary Fig. 4)[40,42]. This indicates that the $Bi^{3+}$ existed in phase-pure $Bi_2S_3$ without other forms. Figure 3d exhibits the corresponding S 2$s$ spectra. It could be split into two subpeaks at 225.4 and 227.6 eV, respectively. The lower-binding-energy peak can be assigned to the chemical bond between Bi and S in $Bi_2S_3$. Another peak, however, is derived from elemental S[40].

During the thermal evaporation, most $Bi_2S_3$ powder was evaporated as the $Bi_2S_3$ molecular form. Meanwhile, partial $Bi_2S_3$ powder was thermally decomposed. Owing to the low vapor pressure of S, both S

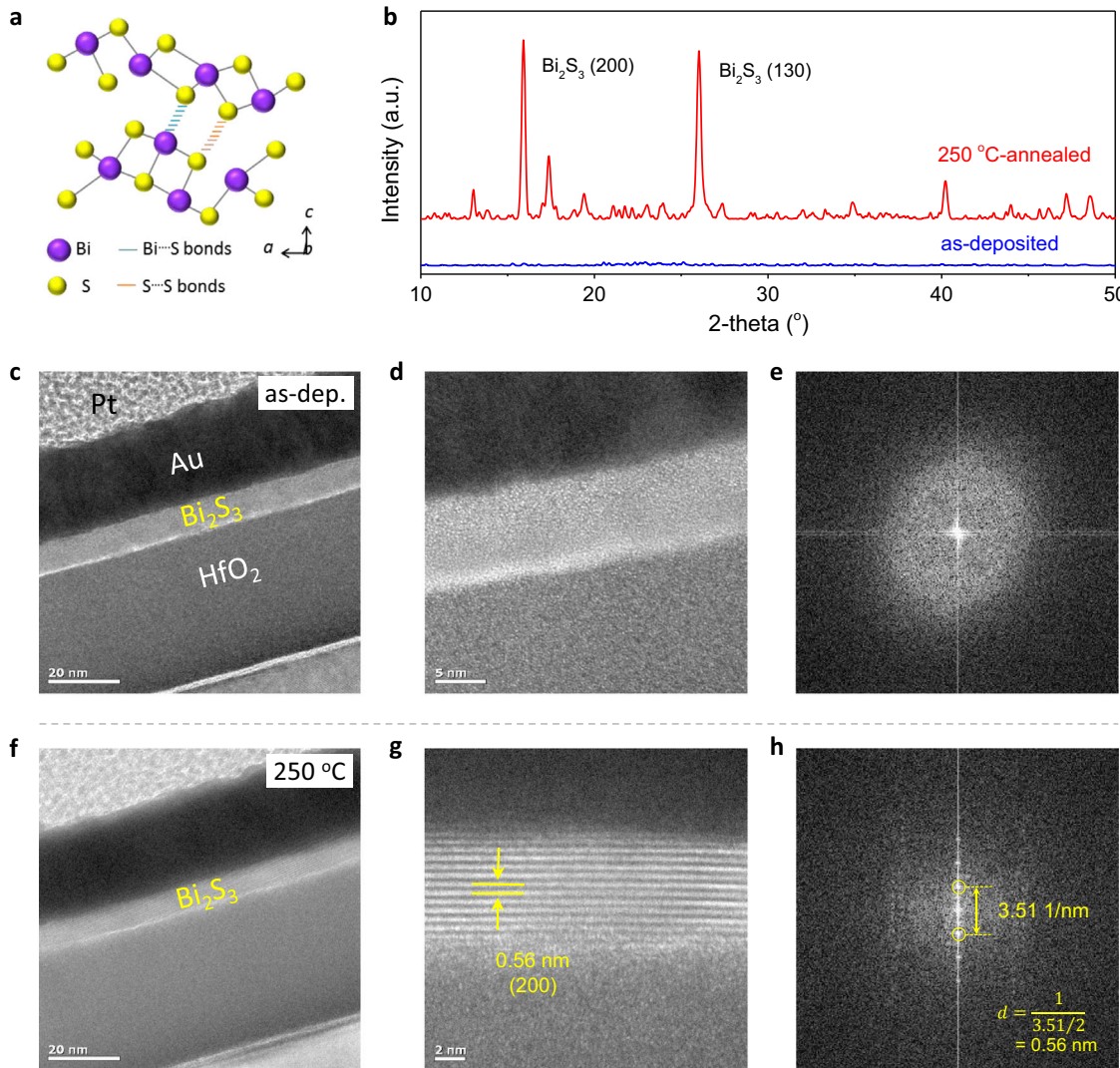

**Fig. 2 | Microstructure analysis of Bi₂S₃ thin films. a** Crystal structure of Bi₂S₃. **b** XRD spectra of as-evaporated Bi₂S₃ thin film and of that annealed at 250 °C. **c–e** HRTEM images and SAED pattern of as-evaporated Bi₂S₃ thin films. **f–h** HRTEM images and SAED pattern of Bi₂S₃ thin film annealed at 250 °C. FFT patterns obtained from the entire area of (**g**). The length of 3.51 1/nm corresponds to a d-spacing of 0.56 nm.

and Bi₂S₃ were evaporated onto the substrates, resulting in the sulfur-rich Bi₂S₃ thin films (Fig. 3e). Residual metallic Bi was observed in the tungsten boat after evaporation (Supplementary Fig. 4). The as-obtained films showed an amorphous structure, resulting from the random distributions of elemental S and Bi₂S₃ molecules. After annealing at 250 °C, the amorphous films turned to crystalline. During annealing, most residual S sublimated, as revealed by the reduced XPS peak intensity. The Bi:S atomic ratio was 2:3.6. Meanwhile, one of the double bonds in Bi₂S₃ was thermally broken, and then the molecules reassembled into crystalline $(Bi_4S_6)_n$ ribbons. This explains the laminar crystalline structure in HRTEM images. Secondary-ion mass spectrometry (SIMS) was used to track the elemental distribution in the films. A uniform Bi distribution was observed in both the samples throughout the bulk. With regard to sulfur, significant enrichment at the bottom was observed in the as-grown Bi₂S₃, which became homogenous after annealing at 250 °C (Supplementary Fig. 6, Fig. 2f).

We then assessed the intrinsic electrical properties of different Bi₂S₃ samples by conducting Hall measurements. The as-grown Bi₂S₃ film showed a high electron concentration of $6.7 \times 10^{19}$ cm⁻³ with a low Hall mobility of ~1 cm² V⁻¹ s⁻¹. The high electron concentration can be attributed to the strong n-doping effect of interstitial sulfur[16]. This is also consistent with the conductor-like behavior of TFTs fabricated

based on the as-evaporated Bi₂S₃ channel. This low mobility can be ascribed to two main factors. One is the strong scattering caused by the high electron concentration and sulfur content. The other is the amorphous state, which generally exhibits a high degree of structural disorder. The electron concentration decreased to $8 \times 10^{15}$ cm² V⁻¹ s⁻¹ after the annealing at 250 °C. The elimination of scattering in conjunction with the enhanced crystalline orientation and film densification contributed to the high Hall mobility of 300 cm² V⁻¹ s⁻¹. It is noteworthy that such extensive electrical property modulation can be achieved conveniently through gentle post-annealing without further processing or doping.

The high mobility remained almost constant even after exposure to air for 30 d, particularly under dry air conditions (Fig. 3g). This indicated remarkable ambient durability. Humid air conditions caused a marginal degradation of mobility. This can exacerbate charge transport owing to moisture absorption at the grain boundaries. This physisorption of moisture is weakly coordinated, and the electrical performance can be recovered rapidly after the baking process at 100 °C. Such ambient stability differs significantly from previous reports on metal chalcogenide films grown by mechanical cleavage or chemical vapor deposition, which show that their electrical properties are sensitive to $O_2$ molecules[43,44]. These film-growth techniques

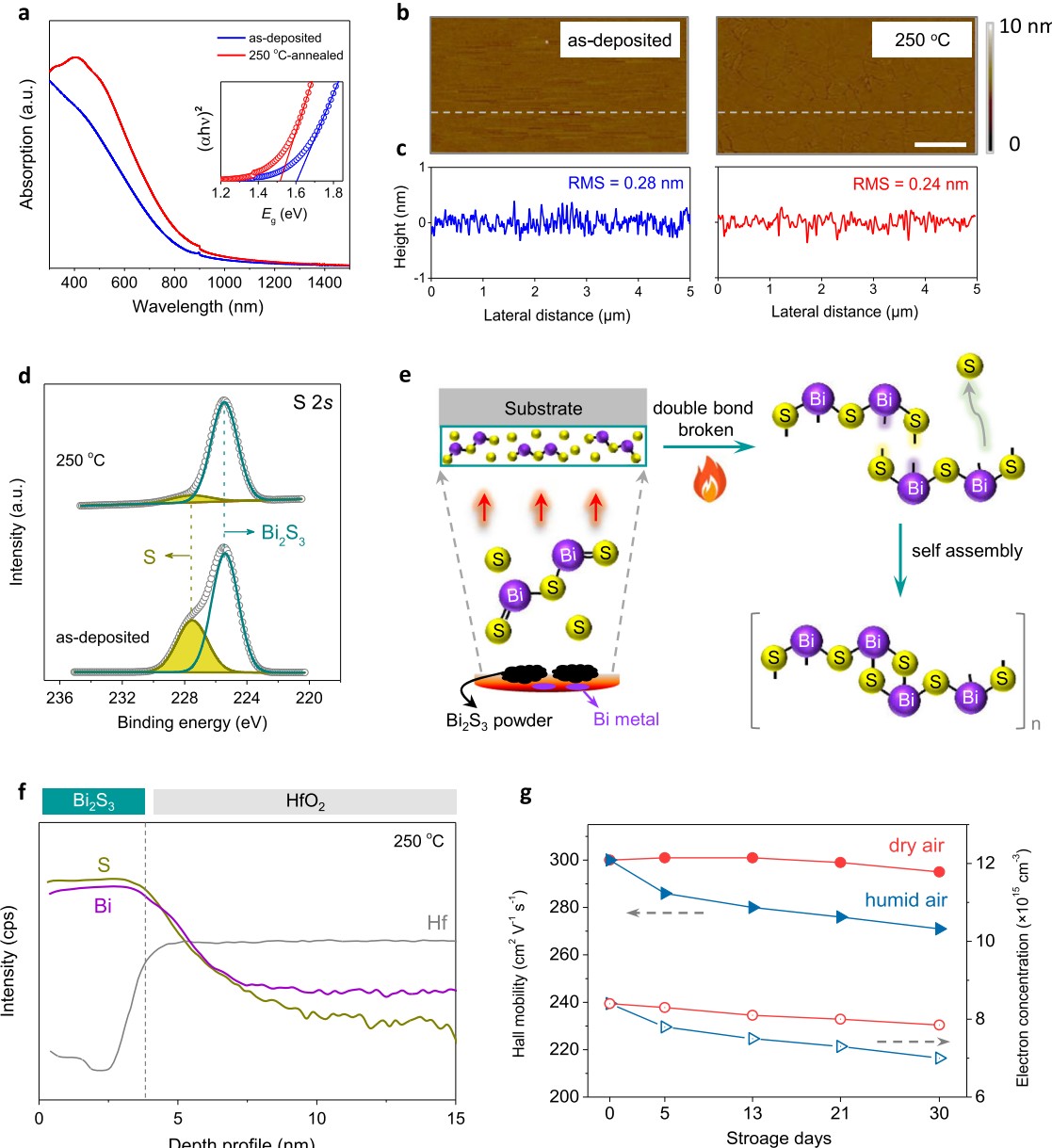

**Fig. 3 | Basic characterizations of evaporated Bi₂S₃ thin films. a** Optical absorption spectra of as-grown and 250 °C-annealed Bi₂S₃ thin films. $E_g$ was calculated by fitting $(\alpha h\nu)^2$ to hν curves using the standard Tauc plot method, as shown in the inset. **b, c** Corresponding AFM images and height profiles (scale bar: 1 μm). **d** Corresponding XPS S 2 s spectra. **e** Schematic of the fabrication of amorphous Bi₂S₃ film composed of sulfur molecules via thermal evaporation. **f** SIMS spectra of 250 °C-Bi₂S₃ thin film. **g** Variations in Hall mobility and electron concentrations of 250 °C-Bi₂S₃ thin films as functions of ambient exposure period and condition (the relative humidity in dry and humid air conditions are <10% and 50–70%, respectively).

generally introduce chalcogen vacancies that induce doping effects through the chemisorption of $O_2$ molecules. For our thermally evaporated Bi₂S₃, the intrinsic marginally sulfur-rich component indicates negligible sulfur vacancies. Thereby, the interaction with $O_2$ reduced. This is supported by the negligible variation in the electron concentration of $(7-8) \times 10^{15}$ cm² V⁻¹ s⁻¹ after long-term air exposure (no doping occurs).

## Electrical characterizations of p-channel Te TFTs and CMOS inverters

We finally explored p-channel devices to realize complementary circuits using thermally evaporated chalcogenide TFTs. The fabrication of high-mobility p-type semiconductors by an inexpensive scalable method is also an urgent task in the electronics community[11,45,46]. Among different candidates, element chalcogenide Te is receiving increased attention owing to its high hole mobility and remarkable stability[23]. Thermal evaporation of Te TFT at a cryogenic temperature of −80 °C was reported to yield a uniform Te channel layer with a large domain size[21]. To enable a more conveniently reproducible film deposition process, we develop the RT thermal evaporation process to deposit Te thin film, which displays a high uniformity, with an RMS value of 0.47 nm (Fig. 4a, b). After mild post-annealing at 100 °C, the Te TFT showed a remarkably high hole mobility ($\mu_h$) of 52 cm² V⁻¹ s⁻¹ and $I_{on}/I_{off}$ of 10⁴ (Fig. 4c, d, Supplementary Fig. 7). The high current linearity and saturation in the output curves indicated Ohmic contact between the Ni electrodes and Te channel (inset in Fig. 4d). Finally, we monolithically integrated CMOS inverters with n-channel Bi₂S₃ and p-channel Te TFTs in a chip. Figure 4e exhibits full-swing characteristics and rapid voltage transitions with a high peak gain of nearly 200 at a supply voltage ($V_{DD}$) of 10 V (Fig. 4f). This indicates the high

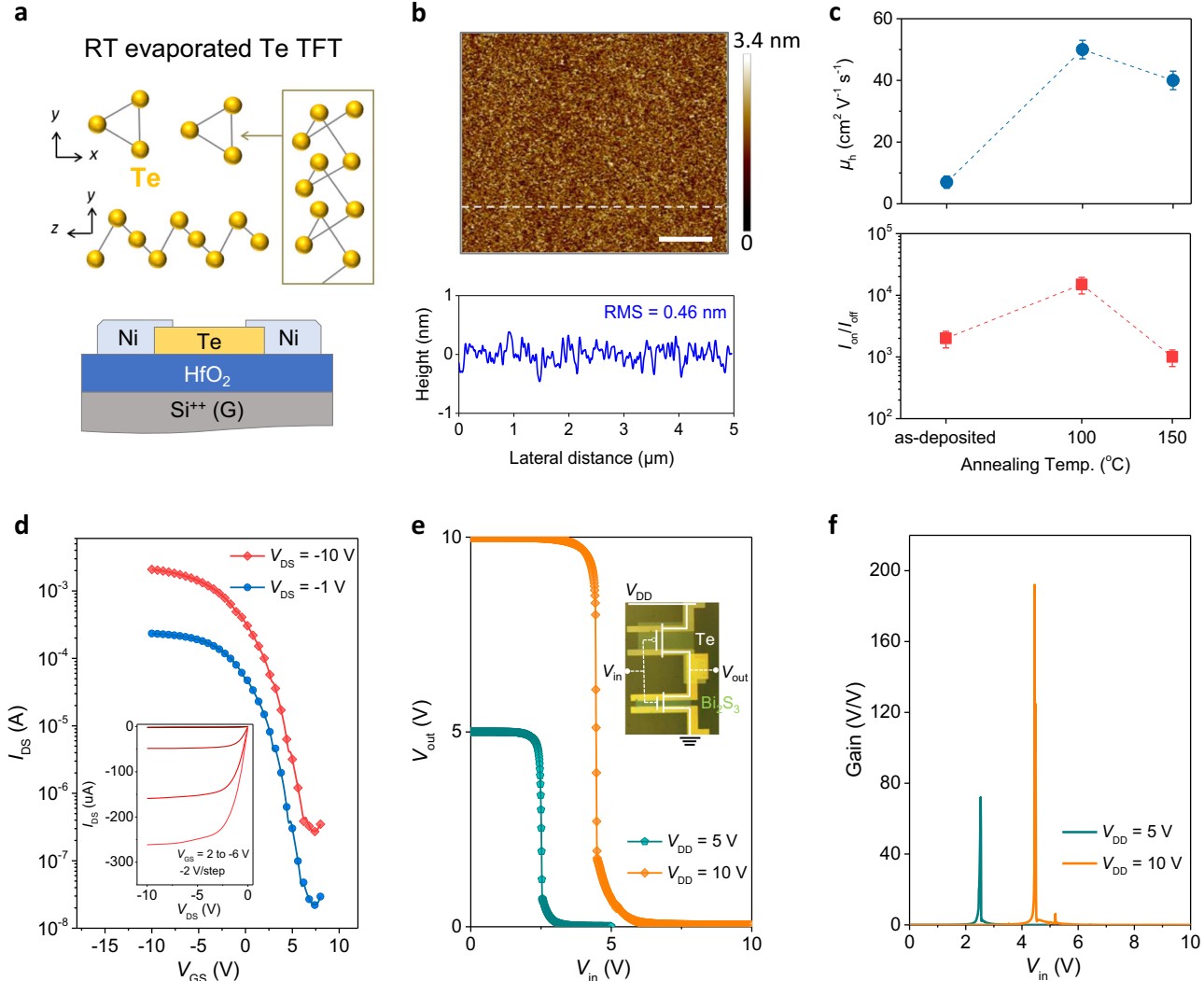

**Fig. 4 | Electrical characterizations of RT evaporated Te TFTs and inverters.**
**a** TFT structure and crystal structure of Te with spiraling chain. **b** AFM image and height profile of as-evaporated Te thin film (scale bar: 1 μm). **c** Summarization of $\mu_h$ and $I_{on}/I_{off}$ of Te TFTs as functions of post-annealing temperature. The error bars were calculated from 10 individual TFTs. **d** Transfer curves of optimized 100 °C-annealed Te TFT measured at linear and saturation regions. The inset shows the corresponding output curves. **e, f** Voltage transfer and gain voltage curves of complementary inverter based on n-channel $Bi_2S_3$ and p-channel Te TFTs at different $V_{DD}$.

potential of thermally evaporated (metal)chalcogenide TFTs for integrating more complex large-area logic circuits.

## Discussion

This work demonstrated the wafer-scale growth of nanometer-scale (metal) chalcogenide thin films and the integration of complementary electronics via standard RT thermal evaporation. Compared with organic semiconductors (which are also compatible with thermal evaporation), (metal)chalcogenides show higher commercial potential as TFT channels because of their higher mobility and stability. Thermal evaporation significantly reduces the use of masks compared with metal oxides deposited by sputtering. In addition to CMOS circuit integration for various applications, thermally evaporated (metal)chalcogenide TFTs provide an opportunity to replace current expensive metal oxide/polycrystalline silicon (LTPO) and pixel-addressing circuits in active-matrix OLED displays[6,47]. This would also enable further integration with thermally evaporated OLEDs in the same chamber and thereby, substantially reduce manufacturing procedures and costs.

The exploration of high-mobility p-type semiconductors capable of large-area deposition in a low-temperature and cost-effective manner

has received substantial attention owing to the highly advanced n-channel TFT technology. Te is becoming an emerging candidate for creating high-performance, stable p-channel transistors[23]. Thermal evaporation provides a simple means to grow scalable, high-quality Te films for laboratory and industrial applications. Our study demonstrates the feasibility of RT growth via the standard thermal evaporation of high-quality Te films. A common issue for Te TFTs is a marginally low $I_{on}/I_{off}$ of $10^4$–$10^5$ with a relatively high OFF current owing to the small $E_g$ of the Te channel (~0.35 eV). In general, an $I_{on}/I_{off} > 10^3$ is feasible for logic circuit operation[7]. However, this results in increased static power consumption. The following are proposed to overcome this issue: (1) $E_g$ enlargement through Se alloying[48] or dimension down to the quantum limit[22] and (2) device engineering through external doping or dielectric encapsulation[49]. In addition, it is worthwhile to consider optimization of the deposition procedures (e.g., substrate temperature, nucleation layer, and deposition rate) and associated film quality in conjunction with contact/dielectric interface engineering, to further improve electrical properties[50–52].

We report wafer-scale growth of nanometer (metal)chalcogenide semiconductors through simple RT thermal evaporation for high-

performance complementary electronics. The n-type $Bi_2S_3$ exhibits unique S-rich-dominated electrical properties with self-assembly crystallization under mild thermal post-annealing conditions. This enables the fabrication of high-performance TFTs with high stability and reproducibility. The combination of high-mobility p-channel Te TFTs further realizes high-gain CMOS inverters. Considering the low vapor pressure and substantially large library of the (metal)chalcogenide family, we anticipate that thermal evaporation would provide a robust and reliable pathway for the scalable production of high-quality functional thin films for large-area and flexible nanoelectronics.

## Methods

### Thin-film fabrication and characterizations

The $Bi_2S_3$ powder (99%) and Te (99.8%) were purchased from Sigma-Aldrich and used directly as evaporation sources. $Bi_2S_3$ and Te films were deposited using the same thermal evaporator via a standard procedure. The substrate temperature was maintained at RT, and the vacuum pressure before evaporation was ~$3 \times 10^{-6}$ Torr. The distance between the substrate and $Bi_2S_3$/Te-loaded tungsten boat was ~20 cm. The thickness of the $Bi_2S_3$/Te film was monitored during deposition. The as-deposited samples were then annealed at different temperatures for 30 min in a $N_2$-filled glove box. The crystal structures of the films on glass were analyzed using XRD with Cu Kα radiation (Bruker D8 ADVANCE). AFM images were obtained using a Nanoscope V Multimode 8 (Bruker, Newark, DE, United States of America) on Si substrates. Optical absorption spectra were obtained using a UV–visible spectrophotometer (V-770, JASCO). Samples for HRTEM characterization were prepared using a focused ion beam (FIB). The images and FFT patterns were obtained using HRTEM (JEOL JEM 2100 F). XPS analysis was performed using a PHI 5000 VersaProbe instrument (Ulvac-PHI, Japan). The depth element distribution was measured by SIMS (IMS 6 F, CAMECA). The Hall measurements of the films were performed in an $N_2$-filled glove box using the van der Pauw method with a 0.51 T magnet at RT.

### TFT fabrication and characterizations

A heavily doped Si wafer (resistivity: 1–100 Ω cm) was used as the substrate and gate electrode. The 40 nm $HfO_2$ grown by ALD at 200 °C was used as the dielectric layer. $Bi_2S_3$ and Te films were deposited on $HfO_2$ as the channel layers, using the procedure described above. The shadow mask was covered with $HfO_2$/Si to obtain patterned $Bi_2S_3$ and Te channel layers for reliable device characterization. Au and Ni source/drain electrodes (40 nm) were deposited on the $Bi_2S_3$ and Te channel layers, respectively, with a shadow mask by using thermal evaporation to construct the bottom-gate, top-contact TFT. The channel length and width (L/W) were 100/800 μm. All the TFTs were characterized at RT in an $N_2$-filled glove box using a Keithley 4200 SCS. The TFT mobilities were calculated in the saturation region using the following equation:

$$\mu_e = \frac{2L}{WC_i}\left(\frac{\partial\sqrt{I_{DS}}}{\partial V_{GS}}\right)^2 \tag{1}$$

$V_{TH}$ was calculated by linearly fitting $I_{DS}^{1/2}$ to $V_{GS}$. $SS$ is the inverse of the maximum slope of the $I_{DS}$–$V_{GS}$ plot.

## Data availability

The data that support the findings of this study are available within the paper and Supplementary Information. Additional relevant data are available from the corresponding authors upon reasonable request.

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

## Acknowledgements

This study was supported by the Ministry of Science and ICT through the National Research Foundation, funded by the Korean Government (NRF-2020M3D1A1110548), Samsung Display Corporation, and LG Display Corporation.

## Author contributions

A.L. and Y.Y.N. conceived the study. A.L. performed the experiments and analyzed the data. H.H.Z. and Y.R. assisted in film characterization and analysis. A.L., T.Y.Z., H.H.Z., and G.S.R. designed the circuit and performed the measurements. A.L. and Y.Y.N. wrote the paper. All the authors have contributed to the final version of this paper.

## Competing interests

The authors declare no competing interests.
