## [Peer Review File · Nature Communications]

Title: Evaporated nanometer chalcogenide films for scalable high-performance complementary electronicsREVIEWER COMMENTS

Reviewer #1 (Remarks to the Author):

The authors report wafer-scale growth of n-type Bi₂S₃ thin films with improved semiconductor characteristics following a room temperature evaporation method and thermal annealing procedure. These n-type films allow for integration with p-type Te based thin films into thin film transistors that are stable and with high reproducibility.

The growth, structural characterization, spectroscopic characterization have all been carried out professionally. The electrical characterizations demonstrating stable on/off ratio, transfer curves etc. reflect sound methodology being followed. The authors provide enough details on the growth and characterization for the work to be reproduced.

I have a few questions that the authors can address:

(1) The fundamental physical process that happens during the annealing process at 250 C - it is not clear if the reduced sulfur vacancies is a result of growth conditions or annealing conditions. A better discussion on that will be helpful to the community.

(2) comparison of transfer characteristics of Bi₂S₃ (this work) thin films with other sulfur-based thin films (MoS₂ etc)

(3) error bars in inset of 1 (f)

In summary, I recommend publication when the questions above are addressed.

Reviewer #2 (Remarks to the Author):

This paper demonstrates wafer scale growth of nanometer scale metal/chalcogenide films. The films are grown at room temperature using thermal evaporation, and a short anneal at 250 C produces material with remarkable mobilities and ultimately devices with typical I_{on}/I_{off} ratios.

I find these results worthy of publication in Nature Communications. The suggested approach is, perhaps, more broadly applicable (i.e. to other metal/chalcogenides). In this respect, the paper will initiate further studies, and device development. In the end, there might be a significant impact on commercial devices.

There is one point that I would hope that the authors could address. While they note that the annealed films are polycrystalline, based on the x-ray data, the authors do not mention a grain size within the film.

In addition, it might be helpful to give a more detailed view of the texture of the polycrystalline films. Such information might provide hints as to how or if the materials performance can be improved. (The higher res TEM images of the lamellar structure suggest that the structure is highly ordered. I am curious if this is "typical" and if so, to understand what is happening during the anneal. This information will also be useful in considering the growth of other metal/chalcogenide films.)

Reviewer #1: The authors report wafer-scale growth of n-type Bi₂S₃ thin films with improved semiconductor characteristics following a room temperature evaporation method and thermal annealing procedure. These n-type films allow for integration with p-type Te based thin films into thin film transistors that are stable and with high reproducibility. The growth, structural characterization, spectroscopic characterization have all been carried out professionally. The electrical characterizations demonstrating stable on/off ratio, transfer curves etc. reflect sound methodology being followed. The authors provide enough details on the growth and characterization for the work to be reproduced. I have a few questions that the authors can address:

Our reply to Reviewer 1:

We thank the reviewer for the comments on our manuscript. We have carefully revised the manuscript based on the reviewer's suggestions, as shown below:

R1Q1. The fundamental physical process that happens during the annealing process at 250 C - it is not clear if the reduced sulfur vacancies is a result of growth conditions or annealing conditions. A better discussion on that will be helpful to the community.

Reply to R1Q1: We thank the reviewer for the good suggestion. The variation of sulfur content is a result of post-annealing conditions.

During thermal evaporation, the Bi₂S₃ powder is mainly evaporated as the molecular form with partial Bi₂S₃ thermal decomposition. Therefore, the as-deposited film contains mainly Bi₂S₃ and elemental sulfur molecules, which is confirmed by XPS analysis (Figure 3d). The existence of elemental sulfur indicates negligible sulfur vacancy in the film. Elemental sulfur acts as the interstitial state, leading to the highly conductive nature. After thermal annealing at 250 °C, the Bi₂S₃ molecules reassembled into the ordered (Bi₄S₆)_n ribbons. Meanwhile, XPS result shows that the majority of sulfur molecules is thermally sublimated, resulting in semiconducting property of Bi₂S₃ thin films.

For more clear discussion and understanding, we reorganized this part in the revised manuscript (page 6).

R1Q2. Comparison of transfer characteristics of Bi₂S₃ (this work) thin films with other sulfur-based thin films (MoS₂ etc).

Reply to R1Q2: We thank the reviewer for the good suggestion.

We compared the transfer characteristics of our Bi₂S₃ TFTs with other large-area metal sulfide-based TFTs from the literature. As summarized in Table R1, the Bi₂S₃ TFT shows higher electrical performance (*i.e.*, I_{on}/I_{off} and mobilities) compared with those other metal sulfide-based TFTs.

One main idea for selecting Bi₂S₃ is its intrinsically high mobility (see Table R2) together with the stable and eco-friendly nature with low cost.

Table R1: Summarizations of key TFT parameters based on different metal sulfides.

Channel materials	Field-effect mobility (cm ² /Vs)	I _{on} /I _{off}	Ref.
Bi ₂ S ₃	12.5	4 × 10 ⁸	this work
MoS ₂	7	10 ⁶	Nature 2018, 562, 256
MoS ₂	10 ⁻²	10 ⁴	Nature Nanotechnology 2021, 16, 592
SnS ₂	2.58	~10 ⁸	Small 2019, 15, 1904116
MnS	0.1	10 ⁶	ACS Nano 2019, 13, 12662
CdS	4.2	~10 ⁸	Sci. Adv. 2018, 4, eaap9104
ReS ₂	7.6	10 ⁴	Adv. Mater. 2016, 28, 6985

Table R2: Summary of the key optoelectronic parameters of different metal sulfides (Nanoscale, 2021, 13, 17272).

Materials	Mobility (cm ² V ⁻¹ S ⁻¹)	Band gap (eV)	CB (eV)	Work function (eV)	Conductivity (S cm ⁻¹)
MoS ₂	200	1.8	4.39	4.42	—
WS ₂	0.1–1	1.36	—	—	1 × 10 ⁻³
TiS ₂	1.38 × 10 ⁻³	1.7	4.02	4.64	7.2 × 10 ⁻⁶
TaS ₂	—	—	4.8	5	—
CdS	4.66	2.4	4.13	4.07	—
SnS ₂	5	2.3	4.24	4.89	7.17 × 10 ⁻⁴
SnS	90	1.3	3.7	4.4	—
In ₂ S ₃	17.6	2.45	3.98	4.32	6.45 × 10 ⁻⁵
Bi ₂ S ₃	257	1.59	3.86	4.45	—
ZnS	3.6 × 10 ⁻⁴	3.4	4.0	—	—
Sb ₂ S ₃	—	1.7	3.7	—	—

R1Q3. Error bars in inset of 1 (f)

Reply to R1Q3: We thank the reviewer for the reminding. We carried out other batches of experiments (tested 4 more TFTs) and added the error bars accordingly in the revised manuscript.

In summary, I recommend publication when the questions above are addressed.

Reviewer #2: This paper demonstrates wafer scale growth of nanometer scale metal/chalcogenide films. The films are grown at room temperature using thermal evaporation, and a short anneal at 250 °C produces material with remarkable mobilities and ultimately devices with typical I_{on}/I_{off} ratios. I find these results worthy of publication in Nature Communications. The suggested approach is, perhaps, more broadly applicable (i.e. to other metal/chalcogenides). In this respect, the paper will initiate further studies, and device development. In the end, there might be a significant impact on commercial devices.

Our reply to Reviewer 2:

We thank the reviewer for the positive comments on our manuscript. We have carefully revised the manuscript, as shown below:

R2Q1. There is one point that I would hope that the authors could address. While they note that the annealed films are polycrystalline, based on the x-ray data, the authors do not mention a grain size within the film. In addition, it might be helpful to give a more detailed view of the texture of the polycrystalline films. Such information might provide hints as to how or if the materials performance can be improved. (The higher res TEM images of the lamellar structure suggest that the structure is highly ordered. I am curious if this is "typical" and if so, to understand what is happening during the anneal. This information will also be useful in considering the growth of other metal/chalcogenide films.)

Reply to R2Q1: We thank the reviewer for the good suggestions.

We provide an enlarged AFM image (Figure R1), where the grain texture of the polycrystalline Bi_2S_3 films can be seen more clearly. The average grain sizes are hundred nanometers. Figure R1 was also added as Figure S3 in the revised manuscript.

For the TEM image, the lamellar structure is typical. We provide a large-scale TEM image (Figure R2). Compared with film grain size, the TEM observation scope is much smaller (a few tens nanometers). Therefore it is reasonable to observe the highly ordered texture in one grain.

For the thermal evaporation of (metal)chalcogenides, the raw material is evaporated as the molecule form. During thermal annealing, the as-evaporated Bi_2S_3 molecules spontaneously reassembled into the order 1D $(\text{Bi}_4\text{S}_6)_n$ ribbons (intrinsic crystalline phase for Bi_2S_3 with lowest formation energy), forming the lamellar texture (illustration in Figure 3e).

The above information and explanations were updated accordingly on Page 6 in the revised manuscript.

Figure R1: AFM image of one 250 °C-annealed Bi₂S₃ film.

Figure R2: HRTEM image of one typical 250 °C-annealed Bi₂S₃ film.

----- End of the response letter.

REVIEWERS' COMMENTS

Reviewer #1 (Remarks to the Author):

The revised manuscript addressed the questions raised by both the referees in a satisfactory manner and I recommend publication.

Reviewer #2 (Remarks to the Author):

The authors have answered all of my questions. I continue to believe that the paper is suitable for publication in Nature Communications.